# PriViT: Vision Transformers for Fast Private Inference

## Abstract

The Vision Transformer (ViT) architecture has emerged as the backbone of choice for state-of-the-art deep models for computer vision applications. However, ViTs are ill-suited for private inference using secure multi-party computation (MPC) protocols, due to the large number of non-polynomial operations (self-attention, feed-forward rectifiers, layer normalization). We propose PriViT, a gradient-based algorithm to selectively "Taylorize" nonlinearities in ViTs while maintaining their prediction accuracy. Our algorithm is conceptually simple, easy to implement, and achieves improved performance over existing approaches for designing MPC-friendly transformer architectures in terms of achieving the Pareto frontier in latency-accuracy. We confirm these improvements via experiments on several standard image classification tasks.

## 1 Introduction

**Motivation.** Deep machine learning models are increasingly being deployed by cloud-based providers, accessible only by API calls. In such cases, user data privacy becomes paramount, motivating the setting of private inference (PI) using secure multiparty computation (MPC). In its simplest form, MPC-based private inference is a two-party setup where a user (the first party) performs inference of their data on a model whose weights are owned by the cloud service provider (the second party), with both sides encrypting their inputs using cryptographic techniques prior to inference.

The main technical barrier to widespread deployment of MPC-based PI protocols is the *large number of nonlinear* operations present in a deep neural network model. Private execution of linear (or low-degree polynomial) operations can be made fast using cryptographic protocols like homomorphic encryption and/or secret sharing. However, private execution of nonlinear operations (such as ReLUs or softmax operations) require Yao's Garbled Circuits, incurring high latency and storage overhead. Thus, unlocking fast, accurate, and efficient PI requires rethinking network design.

Consequently, an emerging line of work has made several forays towards the design of "MPC-friendly" models; cf. more discussions below in Section 2. These methods approach PI from different angles. Approaches such as Delphi (Mishra et al., 2020a) or Circa (Ghodsi et al., 2021) propose to replace ReLUs with MPC-friendly approximations, while approaches such as CryptoNAS (Ghodsi et al., 2020) and Sphynx (Cho et al., 2021) use neural architecture search (NAS) to search for network backbones with a minimal number of ReLUs. Peng et al. (2023) propose hardware-aware ReLU-reduced networks to achieve better latencies. The latest approaches in this direction (SNL by Cho et al. (2022a), and SENet by Kundu et al. (2023)) derive inspiration from network pruning.

However, this body of work has gaps. The overwhelming majority of PI-aware model approaches have focused on convolutional architectures, and have largely ignored *transformer* models. In particular, the proper application of MPC to *vision transformer* architectures remains far less studied; see Table 1. Vision transformers (Dosovitskiy et al., 2020) currently list among the best performing deep models in numerous computer vision tasks, spanning image classification, generation, and understanding. On the other hand, vision transformers are very bulky, possessing an *enormous number of nonlinear operations of different types: GELUs, softmaxes, and layer norms*. As of early September 2023, the only published approach addressing private inference for vision transformers is the MPCViT framework of (Zeng et al., 2022); they use a carefully constructed combination of NAS, various simplifications of the attention mechanism, and knowledge distillation (Hinton et al., 2015) to achieve highly competitive results on common image classification benchmarks.

Table 1: *Comparison of various MPC-friendly approaches for deep image classification. NAS stands for neural architecture search; GD stands for gradient descent. Our approach, PriViT, adaptively replaces various nonlinearities present in transformers with their Taylorized versions in order to reduce PI latency costs without drop in accuracy.*

| Approach | Arch | Methods | Units removed |
|---|---|---|---|
| Delphi (Mishra et al., 2020a) | ConvNets | NAS + poly approx. | ReLU layers |
| CryptoNAS (Ghodsi et al., 2020) | ResNets | NAS | ReLU layers |
| Sphynx (Cho et al., 2021) | ResNets | NAS | ReLU layers |
| DeepReDuce (Jha et al., 2021) | ResNets | manual | ReLU layers |
| SNL (Cho et al., 2022a) | ResNets | GD | Individual ReLUs |
| SENet (Kundu et al., 2023) | ResNets | GD | Individual ReLUs |
| MPCFormer (Li et al., 2022) | BERT | NAS + poly approx. | GELU layers, softmaxes |
| MPCViT (Zeng et al., 2022) | ViT | NAS + poly approx. | GELU layers, softmaxes |
| **PriViT (this paper)** | ViT | GD + poly approx. | Individual GELUs, softmaxes |

**Our contributions and techniques.** In this paper we introduce PRIVIT, an algorithm for designing MPC-friendly vision transformers. PriVit considerably improves upon the previous best results for PI using Vision Transformers (MPCViT) both in terms of latency and accuracy on TinyImagenet, and competitive results on CIFAR 10/100.

At a high level, our approach mirrors the *network linearization* strategy introduced in the SNL method by Cho et al. (2022a). Let us start with a pre-trained ViT model with frozen weights, but now replace nonlinear operations with their *switched Taylorized* versions:

- Each GELU unit, $\text{GELU}(x_i)$ is replaced by $c_i\text{GELU}(x_i) + (1 - c_i)x_i$; and
- Each row-wise softmax operation $\text{Softmax}(X_i)$ is replaced by

$$s_i\text{Softmax}(X_i) + (1 - s_i)\text{SquaredAttn}(X_i).$$

where SquaredAttn is just the unnormalized quadratic kernel, and *binary* switching variables $c_i, s_i$. These switches decide whether to retain the nonlinear operation, or to replace it with its Taylor approximation (linear in the case of GELU, quadratic in the case of softmax[1]). Having defined this new network, we initialize all switch variables to 1, make weights as well as switches trainable, and proceed with training using gradient descent.

Some care needs to be taken to make things work. We seek to eventually set most of the switching variables to zero since our goal is to replace most nonlinearities with linear units or low-degree polynomials; the surviving switches should be set to one. We achieve this by augmenting the standard cross-entropy training loss with a $\ell_1$-penalty term that promotes sparsity in the vector of all switch variables, apply a homotopy-style approach that gradually increases this penalty if sufficient sparsity is not reached, and finally binarize the variables via rounding. We can optionally perform knowledge distillation; see Section 3 for details.

**Discussion and implications.** We note that the previous state-of-the-art, MPCViT, also follows a similar strategy as (Cho et al., 2022a): selectively replace both GELUs and softmax operations in vision transformers with their linear (or polynomial) approximations. However, they achieve this via a fairly complex MPC-aware NAS procedure. A major technical contribution of their work is the identification of a (combinatorial) search space, along with a differentiable objective to optimize over this space. Our PriViT algorithm, on the other hand, is conceptually much simpler and can be applied out-of-the-box to any pre-trained ViT model. The only price to be paid is the computational overhead of training the new switching variables, which incurs extra GPU memory and training time.

While our focus in this paper is sharply on private inference, our results also may hold implications on the *importance of nonlinearities at various transformer layers*. Indeed, we see consistent trends in the architectures obtained via PriViT. First, most nonlinear operations in transformers are redundant. PriViT is able to remove nearly 83% of GELUs and 97% softmax operations with less than 0.5% reduction in accuracy over CIFAR100 (Krizhevsky et al., 2009). Second, given a target overall budget of softmaxes and GELUs, PriViT overwhelmingly chooses to retain most of the nonlinearities in

---

[1]Via several ablation studies we justify why we choose these particular approximations for these functions.

Table 2: *Accuracy-latency tradeoffs between PriVit and MPCViT. All latencies are calculated with the Secretflow Ma et al. (2023) framework using the SEMI2k Cramer et al. (2018) protocol. Detailed methodology is reported in Appendix A* **Left:** *Comparison of PriViT versus MPCViT on TinyImagenet. PriViT achieves* **6.6×** **speedup** *for isoaccuracy approximately 63%.* **Right:** *Comparison of PriVIT versus MPCViT on CIFAR-100. Due to ViT architecture differences, PriViT uses a much larger model with* 3× *more input tokens, and is able to achieve nearly percentage points increase in CIFAR-100 accuracy with only 27% increase in latency. Mirroring the MPCViT+ approach, we also report the effect of PriViT with all GELUs replaced with ReLUs, and again show competitive performance.*

| PriViT | | MPCViT | | PriViT (with GELU) | | MPCViT | | PriVit (with ReLU) | | MPCViT+ | |
|---|---|---|---|---|---|---|---|---|---|---|---|
| Acc | Latency (s) | Acc | Latency (s) | Acc | Latency (s) | Acc | Latency (s) | Acc | Latency (s) | Acc | Latency (s) |
| 78.88 | 31.77 | 62.55 | 150.83 | 78.51 | 17.75 | 77.8 | **9.16** | 78.37 | 16.77 | 77.1 | **9.05** |
| 78.16 | 28.45 | **63.7** | 95.75 | 80.49 | 14.21 | 76.9 | 8.76 | 78.73 | 13.46 | 76.8 | 8.77 |
| 75.5 | 20.47 | 63.36 | 71.04 | 78.5 | 14.08 | 76.9 | 8.21 | 77.1 | 10.62 | 76.3 | 8.37 |
| **64.46** | 14.41 | 62.62 | 43.96 | 77.74 | **11.69** | 76.4 | 7.86 | 76.59 | 12.43 | 76.2 | 7.94 |

earlier layers, while discarding most of the later ones. These suggest that there is considerable room for designing better architectures than merely stacking up identical transformer blocks, but we defer a thorough investigation of this question to future work.

## 2 PRELIMINARIES

**Private inference.** Prior work on private inference (PI) have proposed methods that leverage existing cryptographic primitives for evaluating the output of deep networks. Cryptographic protocols can be categorized by choice of ciphertext computation used for linear and non-linear operations. Operations are computed using some combination of: (1) secret-sharing (SS) (Shamir, 1979; Micali et al., 1987); (2) partial homomorphic encryptions (PHE) (Gentry & Halevi, 2011), which allow limited ciphertext operations (e.g., additions and multiplications), and (3) garbled circuits (GC) (Yao, 1982; 1986).

In this paper, our focus is exclusively on the Delphi protocol (Mishra et al., 2020a) for private inference. We choose Delphi as a matter of convenience; the general trends discovered in our work hold regardless of the encryption protocol, and to validate this we measure latency of our PriViT-derived models using multiple protocols. Delphi assumes the threat model that both parties are honest-but-curious. Therefore, each party strictly follows the protocol, but may try to learn information about the other party's input based on the transcripts they receive from the protocol. Wang et al. (2022), Peng et al. (2023), Lu et al. (2021), Qin et al. (2022)

Delphi is a hybrid protocol that combines cryptographic primitives such as secret sharing (SS) and homomorphic encryptions (HE) for all linear operations, and garbled circuits (GC) for ReLU operations. Delphi divides the inference into two phases to make the private inference happen: the offline phase and an online phase. Delphi's cryptographic protocol allows for front-loading all input-independent computations to an offline phase. By doing so, this enables ciphertext linear computations to be as fast as plaintext linear computations while performing the actual inference. For convolutional architectures, the authors of Delphi shows empirical evidence that ReLU computation requires 90% of the overall private inference time for typical deep networks. As a remedy, Delphi and SAFENET (Lou et al., 2021) propose neural architecture search (NAS) to selectively replace ReLUs with polynomial operations. CryptoNAS (Ghodsi et al., 2020), Sphynx (Cho et al., 2021) and DeepReDuce (Jha et al., 2021) design new ReLU efficient architectures by using macro-search NAS, micro-search NAS and multi-step optimization respectively.

**Protocols for nonlinearities.** To standardize across different types of non-linear activations, we compare their Delphi (online) GC computation costs. We use the EMP Toolkit (Wang et al., 2016), a widely used GC framework, to generate GC circuits for nonlinear functions. High-performance GC constructions implement AND and XOR gates, where XOR is implemented using FreeXOR (Kolesnikov & Schneider, 2008) and AND using Half-Gate (Zahur et al., 2015). With FreeXOR, all XOR gates are negligible, therefore we count the number of AND gates as the cost of each nonlinear function (Mo et al., 2023b). To be consistent with prior work (Ghodsi et al., 2021), the activation functions also consider value recovery from Secret Sharing. Figure 1 (left) breaks down the GC cost of ViT for different nonlinearities, and (right) shows the # AND gates in Softmax and GeLU. Figure 2 breaks down softmax into fundamental operations, these operations are already

synthesized and included in the EMP Toolkit library. Thus we simply add all the AND gates of these basic operations to arrive at the total number of AND gates of softmax operations.

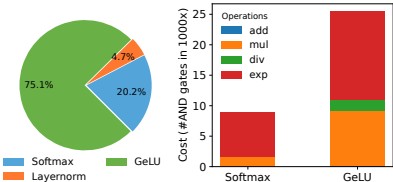 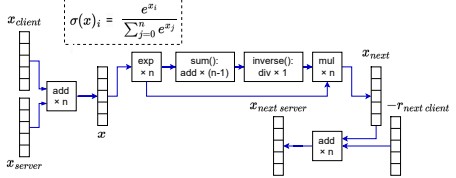

Figure 1: *Breakdown of latency in ViT-Tiny model of different non-linearities based on Delphi.*

Figure 2: *Detailed steps of benchmarking the non-linearity cost for softmax.*

# 3 PRIVIT: PRIVACY FRIENDLY VISION TRANSFORMERS

## 3.1 SETUP

Following (Cho et al., 2022a; Ghodsi et al., 2020; Mo et al., 2023a), we exclusively focus on Delphi (Mishra et al., 2020b) as the protocol for private inference. However, we emphasize this choice is only due to convenience, and that our approach extends to any privacy-preserving protocol that relies on reducing nonlinearities to improve PI latency times.

Let $f_{\mathbf{W}} : \mathbb{R}^{n \times d} \to [0, 1]^C$ be a vision transformer that takes as input $n$ tokens (each of $d$ dimensions) and outputs a vector of probabilities for each of $C$ classes. Each of these tokens is a patch sampled from the original image, $\mathbf{X}$ and is indexed by $i$. As described, the transformer architecture consists of stacked layers of multi-headed self-attention blocks with nonlinearities like GeLU (Hendrycks & Gimpel, 2016) and Layernorm (Ba et al., 2016). ViTs use dot-product self-attention (see Equation 1) which additionally consists of $n$ row-wise softmax operations.

$$o = \frac{\text{Softmax}(\mathbf{X}\mathbf{W}_q\mathbf{W}_k^T\mathbf{X}^T)}{\sqrt{d}}\mathbf{X}\mathbf{W}_v. \tag{1}$$

To frame the computational challenges inherent to Vision Transformers (ViTs), consider the ViT-base (12 layer) model designed for $224 \times 224$ images. Delving into its architecture reveals a composition of (approximately) $726,000$ GeLUs, $28,000$ softmax, and $4000$ layer norms. All the non-linearities, when viewed through the lens of the Delphi protocol, become extremely resource-intensive operations.

Our PriViT algorithm designs an architecture that circumvents these computationally heavy operations. Our proposition is to surgically introduce appropriate Taylor approximations of the GeLU and softmax attention operations wherever possible (under the constraint that accuracy drops due to such approximations should be minimal. The main challenge is to figure out where to do these approximations, which we describe below,

Our algorithm can be viewed as an extension of SNL (Cho et al., 2022b), a network linearization approach. SNL allows for automatic linearization of feed-forward networks through the use of parametric ReLU activations and optimizing a Lasso-like loss (Tibshirani, 1996). While SNL can reasonably be used to linearize ReLUs (GeLUs) in ViTs, it does not support linearizing softmax operations, which form a large proportion of nonlinearities in ViTs. We therefore add a reparametrized normalization layer that allows a choice between softmax and SQUAREDATTN. Note that this is distinct to many existing approaches (Qin et al., 2022; Lu et al., 2021; Wang et al., 2020; Song, 2021) which also propose *blanket* alternatives to softmax attention throughout the network.

## 3.2 PRIVIT ALGORITHM

To begin, we focus on softmax and GeLUs and ignore layernorms; we found that these were far harder to Taylorize. For the former, we introduce auxiliary variables to act as switches. Given $f_{\mathbf{W}}$, let $\overline{C}$ and $\overline{S}$ be the total number of GeLUs and softmaxes. Further, let $\mathcal{S} = [s_1, s_2, ..., s_S]$ and $\mathcal{C} = [c_1, c_2, \ldots, c_G]$ be collections of binary switch variables defined for all instances of GeLU and softmax activations. Our goal here is to learn $\mathbf{W}, \mathcal{S}$, and $\mathcal{C}$ to ensure high accuracy with as few

nonlinearities as possible. We also use $N$ to denote the number of tokens, $H$ to denote the number of heads and $m$ to denote the size of the token embedding (and consequently the output size of the feedforward MLP).

**GELU.** In the case of GELU operations, we define a switched version of the GeLU activation:

$$f(c_i, \mathbf{x}_i) = c_i \text{GELU}(\mathbf{x}_i) + (1 - c_i)\mathbf{x}_i \tag{2}$$

$$\mathbf{y} = \big[ f(c_1, x_1), f(c_2, x_2), \ldots, f(c_n, x_n) \big], \tag{3}$$

where $c_i$ is the corresponding auxiliary variable for the $i^{\text{th}}$ token, $\mathbf{x}_i$ is the $i^{\text{th}}$ input token embedding of dimension $m$ ($m$ being the MLP dimension) and $\mathbf{y} \in \mathbb{R}^{N \times m}$ is the output. During training, $c_i$ are initially real-valued, trainable, and are initialized to 1 at the start of training. During inference, we binarize all $c_i$ using an indicator function, $\mathbb{1}_{c_i > \epsilon}$, where $\epsilon$ is an appropriately chosen threshold. $c_i = 1$ implies that the GELU is preserved whereas $c_i = 0$ reverts to the linear activation. Figure 13 in the Appendix shows a graphical representation of the GELU parametrization. Note that GELU is a pointwise function and therefore is applied to elementwise.

**Softmax Attention.** The next step is to reparameterize softmax attention. However unlike GELUs, choice of parameterization is not obvious here. As per the Delphi protocol, exponents are extremely expensive to calculate. On the other hand, polynomials are comparatively cheaper. Also division by a constant can be folded away compared to division by a number that is input dependent as in the case of softmax. Therefore, we propose a modified 'Squared Attention' block;

$$\text{SQUAREDATTN}(\mathbf{X}) = \frac{\big(\mathbf{X}\mathbf{W}_q\mathbf{W}_k^T\mathbf{X}^T\big)^2}{N} \mathbf{X}\mathbf{W}_v, \tag{4}$$

wherein we apply pointwise squaring instead of a row-wise softmax and divide by the number of tokens. Squared attention is MPC friendly for the properties described above, all the while preserving performance compared to original softmax. Similar to our approach with GELUs, we further add a learnable auxiliary variable, $s_i$ for every row-wise softmax operation in the attention layer.

$$o = s_i \text{Softmax}(\mathbf{X}_i) + (1 - s_i)\text{SQUAREDATTN}(\mathbf{X}_i), \tag{5}$$

where $\mathbf{X}_i$ is the $i^{\text{th}}$ row of the attention matrix. As before, $s_i$s are initially real-valued, trainable and initialized to 1. The variables are binarized during inference allowing use of either Softmax or squared attention based on the values of $s_i$. Further ablations of different candidate attention functions are presented in the results sections.

### 3.3 TRAINING PRIVIT

To train PriVit models, we need to train three sets of variables: the weights of the transformer, $\mathbf{W}$, the switch variables for the GELU parameterization, $\mathcal{C}$, and the switch variables for the attention parametrization, $\mathcal{S}$. Our goal is to train a model that minimizes the number of nonlinearities to satisfy a given nonlinearity budget, that is, $\|\mathcal{C}\|_0 < C$, and $\|\mathcal{S}\|_0 < S$, while increasing the overall performance. This is reminiscent of standard LASSO-style (Tibshirani, 1996) optimization. We therefore propose the following loss function to train the model,

$$L_{privit} = L(f_\mathbf{W}(\mathbf{X}), y) + \lambda_g \sum_{i=0}^{|\mathcal{G}|} |c_i| + \lambda_s \sum_{j=0}^{|\mathcal{S}|} |s_i|, \tag{6}$$

where $L$ is the standard cross-entropy loss. We then optimize for each of the variables until the required softmax attention and GELU budgets. We show pseudocode for our training algorithm in Algorithm 1 in the Appendix.

Optionally, we can also make use of knowledge distillation during both training and fine-tuning. We introduce a KL divergence loss on the soft labels generated by the teacher and student ViT model. This loss is added to the $L_{privit}$ loss defined in eq. 6. Thus our final minimization objective looks as follows,

$$\min_{\mathbf{W}, \mathcal{C}, \mathcal{S}} L(f_\mathbf{W}(\mathbf{X}), y) + \lambda_g \sum_{i=0}^{|\mathcal{G}|} |c_i| + \lambda_s \sum_{j=0}^{|\mathcal{S}|} |s_i| + L_{kl}(f_\mathbf{W}(\mathbf{X}), f_\mathbf{T}(\mathbf{X})) \tag{7}$$

where $\mathbf{T}$ denotes the weights of the teacher model, and $L_{kl}$ is the KL divergence loss. After every epoch, we count the number of GELUs and softmax attention operations by thresholding the $s_i$ and $c_i$ values. Once the model satisfies the required budgets, we freeze the chosen GELUs and softmax attention operations by binarizing all $s_i$ and $c_i$ values and fine-tune the model weights for the classification task. Figure 15 provides a complete illustration.

# 4 RESULTS

## 4.1 EXPERIMENTAL SETUP

**Architecture and dataset**. We apply PriViT algorithm to a pretrained checkpoint of ViT-Tiny(Steiner et al., 2021) that is trained on ImageNet-21k (14 million images, 21,843 classes) at resolution $224 \times 224$, and fine-tuned on ImageNet 2012 (1 million images, 1,000 classes) at resolution $224 \times 224$. The pretrained ViT Tiny checkpoints are made available by (WinKawaks, 2022). In this research work we focus on finetuning an existing model checkpoint like ViT Tiny on a target standard image classification dataset (CIFAR10/100 (Krizhevsky et al., 2009) and Tiny-ImageNet). CIFAR10/100 has images of size $32 \times 32$ while Tiny-ImageNet has $64 \times 64$. These images were resized to $224 \times 224$ before being given as an input. CIFAR10 has 10 classes with 5000 training images and 1000 test images per class. CIFAR100 has 100 classes with 500 training images and 100 test images per class. Tiny-ImageNet has 200 classes with 500 training images and 50 test images per class. We also perform hyperparameter tuning and present more details in Appendix A

**ViT teacher pretraining**. As the base model, we finetune a pretrained ViT-Tiny on CIFAR10/100 for 10 epochs. We use AdamW (Loshchilov & Hutter, 2017) as the optimizer with an initial learning rate and weight decay as 0.0001 and 0.0001 respectively, and decay the learning rate after every 30 epochs by multiplying it by 0.1. Batch size used is 64. We use the same hyperparameters for the TinyImagenet model as well. We use these weights to initialize PriViT and start KD.

**Joint optimization of student ViT and parametric non linearities**. We use Adam (Kingma & Ba, 2014) optimizer with learning rate equal to 0.0001. We use knowledge distillation and use soft labels generated by the teacher model with a temperature of 4. The total loss is then, $L = L_{\mathrm{PriViT}} + L_{\mathrm{KL}}$, where $L_{\mathrm{PriViT}}$ is Equation 6 and $L_{\mathrm{KL}}$ is the KL divergence loss between the logits of teacher and student model. The Lasso coefficient (Tibshirani, 1996) for parametric attention and GELU mask are set to $\lambda_g = 0.00003$ and $\lambda_s = 0.00003$ respectively at the beginning of the search. We set warmup epochs to 5 during which we don't change any hyperparameters of the model. Post warmup, we increment $\lambda_g$ by a multiplicative factor of 1.1 at the end of each epoch if the number of active GELUs of current epoch do not decrease by atleast 2 as compared to previous epoch. Note that a GELU/softmax is considered active if it's corresponding auxiliary variable is greater than threshold hyperparameter $\epsilon = 0.001$. We follow the same approach for $\lambda_s$, with a multiplicative factor of 1.1 and an active threshold of 200.

**Binarizing parametric nonlinearities, finetuning.** When the GELUs and softmax budgets are satisfied, we binarize and freeze the the GELU and softmax auxiliary variables. We subsequently finetune the model for 50 epochs using AdamW with a learning rate 0.0001, weight decay 0.0001 and a cosine annealing learning rate scheduler (Loshchilov & Hutter, 2016). Our finetuning approach continues to use knowledge distillation as before.

**Non-linearity cost comparison.** We conduct experiments to assess the computational cost of non-linear functions such as layernorm, softmax, and GeLU in comparison to ReLU within GC. The detailed results are reported in Table 7, brackets considers amortizing to a vector of inputs, e.g., a Layernorm(192) is an operation over a vector length of 192 is equivalent to $6504\times$ than the cost of a ReLU. It demonstrates that with a vector length of 197, all layernorm and softmax functions incur higher computational costs (i.e., number of ANDs) than ReLU. Specifically, they exhibit costs $6504\times$, $18586\times$ higher than that of ReLU respectively and for pointwise GELU, we saw a cost $270\times$ higher than that of ReLU. The cost of denominator of layernorm and softmax can be amortized to the whole vector and thus incur less cost than GELU. We estimate the latency of each model generated by PriViT using these conversion factors. To show an example, we estimate the non-linearity cost of a hypothetical model with 1000 softmax operation, 1000 layernorm operations and 1000 GELUs, by taking the weighted sum of each operations with their corresponding latency factor.

Table 3: *Base model architecture of PriViT and MPCViT*

| Model | Layers | Width | MLP | Heads | Image size | Patch size | params (M) |
|---|---|---|---|---|---|---|---|
| PriViT | 12 | 192 | 768 | 3 | 224×224 | 16×16 | 5.8 |
| MPCViT (Tiny Imagenet) | 9 | 192 | 384 | 12 | 64×64 | 4×4 | - |
| MPCViT (Cifar 10/100) | 7 | 256 | 512 | 4 | 32×32 | 4×4 | 3.72 |

Table 4: *Comparison of PriViT-R, PriViT-G, and MPCViT over Tiny Imagenet*

| PriViT - R | | PriViT - G | | MPCViT | |
|---|---|---|---|---|---|
| Accuracy | Latency (M) | Accuracy | Latency (M) | Accuracy | Latency (M) |
| **64.73** | 69.11 | 69.8 | 151.75 | 62.55 | 381.42 |
| 61.05 | 67.08 | 66.98 | 128.23 | **63.7** | 331.35 |
| 56.83 | 69.28 | **64.46** | 110.60 | 63.36 | 307.45 |
| 57.65 | 82.77 | 60.53 | 93.72 | 62.62 | 282.42 |

Table 5: *Comparison of training efficiency between PriViT and MPCViT on ViT-Tiny*

| Dataset | PriViT | | | MPCViT | | |
|---|---|---|---|---|---|---|
| | Latency | Accuracy | Epochs | Latency | Accuracy | Epochs |
| TinyImagenet | 151.75 | 69.8 | 293 | 381.42 | 62.55 | 600 |
| | 128.23 | 66.98 | 351 | 331.35 | 63.7 | 600 |
| | 110.60 | 64.46 | 342 | 307.45 | 63.36 | 600 |
| CIFAR 100 | 88.24 | 78.5 | 403 | 72.21 | 77.8 | 900 |
| | 75.92 | 77.74 | 447 | 71.77 | 76.9 | 900 |
| | 67.54 | 75.47 | 498 | 71.40 | 76.9 | 900 |

**GELU replacement post training.** Mirroring the MPCViT+ approach, we also report the effect of PriViT with all GELUs replaced with ReLUs, we call this model PriViT - R, and the original PriViT model as PriViT - G. It is important to note that such an optimization is effective in low GELU budgets as it introduces very minimal errors. In high GELU budgets the error is quite significant that it affects the overall performance.

## 4.2 COMPARISON WITH PRIOR ART

We benchmark PriViT against MPCViT, using the checkpoints publicly shared by the authors. We use the latency estimates reported in Section 4.1 and report the total latency. Specifically we convert latency contribution of non-linear operations to *ReLU* equivalents. We refer to the latency of a single RELU operation as 'RELUOps' for a given system. We can therefore measure other non-linearities in terms of RELUOps. This proxy has the advantage that it abstracts away system level variables like hardware, memory and bandwidth which often cause variance in bench marking performance. Table 3 highlights the differences in base model architecture of PriViT and MPCViT.

**Pareto analysis of PriViT over TinyImagenet, and Cifar10/100.** In our evaluation on various datasets, the performance of PriVit was benchmarked against both MPCViT and MPCViT+. We measure two metrics of importance – the latency (measured in terms of RELUOps), and accuracy. An ideal private inference algorithm will achieve high accuracy with low latency.

1. **Tiny ImageNet**: Using a Pareto analysis on the Tiny ImageNet dataset, PriViT showcases notable improvement. On Tiny imagenet, for an isoaccuracy of approximately 63% , PriViT G and PriViT R achieved 3× and 4.7× speedup compared to MPCViT respectively as reported in Table 4.
2. **CIFAR-10**: We observe from our results in Figure 3 that in certain latency regimes PriVit performs just as well as MPCViT and slightly worse than MPCViT+ in the trade-off between performance and computational efficiency.
3. **CIFAR-100**: Turning our attention to the CIFAR-100 dataset, the performance nuances became more evident. PriViT G performs just as well as MPCViT but is slightly worse compared to MPCViT+. However, when benchmarked against PriViT R, PriVit's performance was much better than MPCViT and MPCViT+, indicating the competitive nature of the two algorithms on this dataset.

Table 5 shows that PriViT, at a similar accuracy ( 64%), requires about half the training epochs compared to MPCViT on TinyImagenet. For an isolatency of 75M on CIFAR 100, PriViT also needs only about 50% of MPCViT's training epochs. This demonstrates PriViT's enhanced efficiency and scalability, making it a promising alternative to MPCViT, particularly in situations valuing efficiency and performance.

## 4.3 ABLATION STUDIES

**Contribution by Knowledge Distillation.** In PriViT, we incorporate knowledge distillation (KD) alongside supervised learning. To assess the contribution of KD to the overall performance, we trained PriViT on the TinyImagenet dataset with varying non-linearity budgets. We then compared its

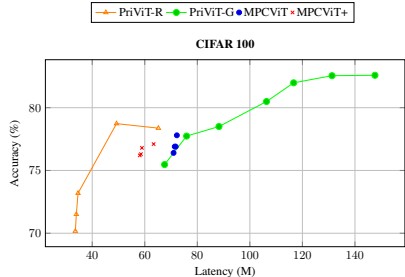 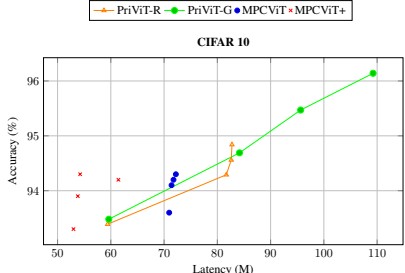

Figure 3: Comparison of PriViT over CIFAR 10/100 benchmarked against MPCViT, and MPCViT+. The latency is calculated as per Section 4.1

Table 6: *Latency comparison between PriViT and PriViT w/o Pretrain*

| PriViT | | PriViT w/o pretrain | |
|---|---|---|---|
| Latency (M) | Accuracy (%) | Latency (M) | Accuracy (%) |
| 271.59 | 75.5 | 234.18 | 53.57 |
| **151.74** | 69.8 | 194.78 | 54.66 |
| 128.23 | 66.98 | 167.20 | 55.59 |
| 93.71 | 60.53 | **153.31** | 55.92 |

Table 7: *Non-linearity cost normalized to the cost of one ReluOp which is 1 ReLU operation over a scalar value.*

| PriViT | | MPCViT TinyImagenet | | MPCViT CIFAR10/100 | |
|---|---|---|---|---|---|
| Function | # ReluOps | Function | # ReluOps | Function | # ReluOps |
| Softmax(197) | 18586 | ReLU Softmax(257) | 4428 | ReLU Softmax(65) | 1133 |
| Layernorm(192) | 6504 | Layernorm(192) | 6504 | Layernorm(256) | 8614 |
| GeLU(1) | 270 | GeLU(1) | 270 | GeLU(1) | 270 |
| $x^2$(197) | 3248 | | | | |

performance to a version of PriViT (as outlined in Figure 15) that does not employ a teacher model for knowledge distillation. Our results in figure 4 indicate that, under identical latency conditions, incorporating KD enhances performance by approximately 5%.

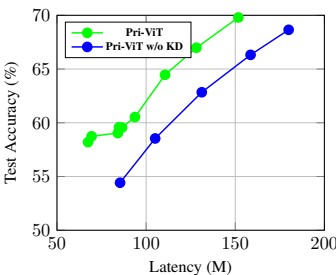 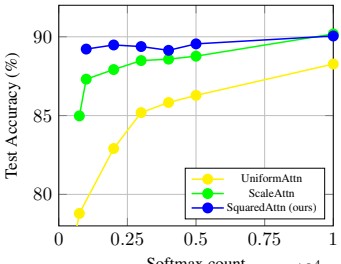

Figure 4: *We evaluated PriViT with and without KD. The x axis represents latency measured as per Section 4.1, while the y axis shows the accuracy on TinyImagenet. We observe an overall improvement in the latency-accuracy curve motivating the use of KD.*

Figure 5: *We evaluated the PriViT algorithm using three attention operations: Uniform, Linear, and Squared Attention. The x-axis represents the target softmax count, while the y-axis shows the test accuracy on CIFAR100.*

**Contribution of pretraining.** In PriViT, we utilize a pretrained checkpoint, which is subsequently fine-tuned. Post fine-tuning, we introduce a parametric GeLU and attention mechanisms to decrease non-linearities in the model. To gauge the impact of using a pretrained model on the overall performance, we contrast the performance of PriViT with a variant of PriViT that is not built upon a pretrained model. Instead, this variant employs weights initialized from scratch and is trained with the same parametric non-linearity mask as used in PriViT to minimize non-linearities. The comparative outcomes of these approaches are presented in Table 6. Our findings reveal that, for comparable latencies, PriViT with the pretrained checkpoint outperforms its counterpart without it, registering a 14% enhancement in accuracy.

**Choice of softmax approximation.** To highlight the contribution of different attention candidate, we run PriViT over different softmax budget over CIFAR100, and report the accuracy of the resulting model versus the number of original softmax attention retained. Lower number of softmax operations implies higher the number of softmax attention replaced with our candidate attention operation. As per Figure 5 we see almost no performance drop for SQUAREDATTN, roughly 5% drop in performance for SCALEATTN and 10% drop in performance for UNIFORMATTN in low budgets.

Table 8: *Comparing PriViT and layerwise linearization of GeLU in a ViT model with 200k GeLUs. Six models were generated by replacing two GeLU layers at a time with Identity.*

| Layerwise GELU linearizing | | Pri-ViT | |
|---|---|---|---|
| Gelu (K) | Acc (%) | Gelu (K) | Acc (%) |
| 197 | 96.07 | 200 | 95.59 |
| 193 | 95.91 | 150 | 95.34 |
| 187 | 94.28 | 100 | 95.58 |
| 181 | 93.33 | 50 | 94.98 |
| 174 | 93.04 | 10 | 94.24 |
| 164 | 92.06 | 1 | 93.96 |
| 123 | 82.48 | | |
| 0 | 56.64 | | |

Thus SquaredAttention outperformed the others across all softmax budgets, motivating its selection to replace the standard softmax attention in PriViT.

**Fine-grained versus layer-wise Taylorization** PriVit employs a unique approach where it selectively Taylorizes softmax and GELU operations. To probe the effectiveness of this method, we contrasted it with an alternative PriViT approach that Taylorizes a ViT model progressively, layer by layer. As illustrated in Table 8, our observations underscored the superiority of selective Taylorization.

**Visualization of non-linearity distribution.** To understand which nonlinearities are preserved, we investigate the distribution of PriViT models under different softmax and GELU budgets. From our observations in Figure 6 we can conclude that GELUs in earlier encoder layers are preferred over the ones in the later layers. From figure 7 we observe a similar trend in softmax distributions. We find this interesting, since the trends reported in earlier work on convolutional networks are in the reverse direction: earlier layers tend to have a larger number of linearized units. Understanding this discrepancy is an interesting question for future work.

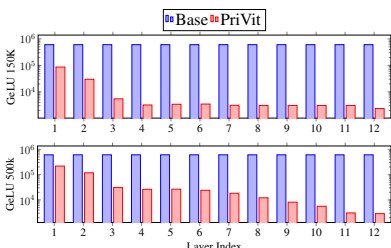

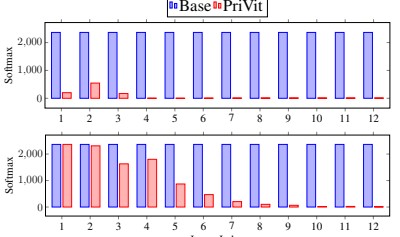

Figure 6: *Comparison of GELU distribution between ViT-base (Base) and PriViT without softmax linearization. The x-axis represents the model's layer index, while the y-axis shows log-scaled GELU operations per layer. With an input tensor size of $197 \times 3072$ for the GELU layer, each layer contains $197 \times 3072 = 605184$ GELU operations. **Top**: 150K target GELU. **Bottom**: 500K target GELU.*

Figure 7: *Comparison of softmax distribution in ViT-base model (Base) versus PriViT without GeLU linearization. The $x$-axis denotes the layer index, while the $y$-axis shows the softmax operations per layer. With a $197 \times 197$ attention matrix across 12 heads, the ViT-base model totals 2364 softmax operations per layer. Notably, PriViT tends to substitute earlier layer softmaxes with linear operations. **Top**: 1K target softmax; **Bottom**: 10K target softmax.*

## 5 CONCLUSION

We introduce PriViT, a new algorithm for designing MPC-friendly vision transformers, and showed its competitive performance on several image classification benchmarks. A natural direction of future work is to extend similar techniques for designing other families of transformer architectures, such as Swin Transformers and Data Efficient image transformers (DEiT), as well as encoder-decoder transformer architectures. A key limitation of PriViT is it's inability to Taylorize layernorms without introducing instability in the training.

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

Table 9: Benchmarking PriViT and MPCViT over CIFAR 10 dataset on SEMI2k protocol.

| PriViT G | | MPCViT | | PriVit R | | MPCViT+ | |
|---|---|---|---|---|---|---|---|
| Acc | Latency (s) | Acc | Latency (s) | Acc | Latency (s) | Acc | Latency (s) |
| 96.31 | 21.13 | 94.3 | 10.39 | 94.45 | 18.74 | 93.3 | 5.57 |
| 95.31 | 19.27 | 94.2 | 9.85 | 92.45 | 17.83 | 93.9 | 6.38 |
| 95.58 | 15.99 | 94.1 | 9.39 | 92.36 | 13.26 | 94.2 | 7.39 |
| 95.14 | 14.43 | 93.6 | 8.96 | 92.39 | 13.01 | 94.3 | 6.83 |
| 94.52 | 14.37 | | | 91.08 | 10.24 | | |
| 94.44 | 11.6 | | | | | | |

## A    SUPPLEMENTARY RESULTS

**Additional PI results.** Following Zeng et al. (2022) we benchmark our method over SEMI2k using secretflow framework, the client and server are 64GB RAM, Intel(R) Xeon(R) Platinum 8268 CPU @ 2.90GHz. We run PI over LAN settings between two nodes of HPC cluster, hence there is a variation of the total inference latency from what is reported in Zeng et al. (2022), but to keep a consistent comparison, we benchmark both PriViT and MPCViT under our system settings. We report additional bench marking results on CIFAR 10 data in the Table 9.

**PriViT on different architectures.** Figure 8 provides a comparison of PriViT over different ViT variants from 5M (ViT-Tiny) to 80M (ViT-Base) parameters. We see that PriViT is able to produce a similar latency accuracy trade off across different model sizes emphasizing it's generalizability.

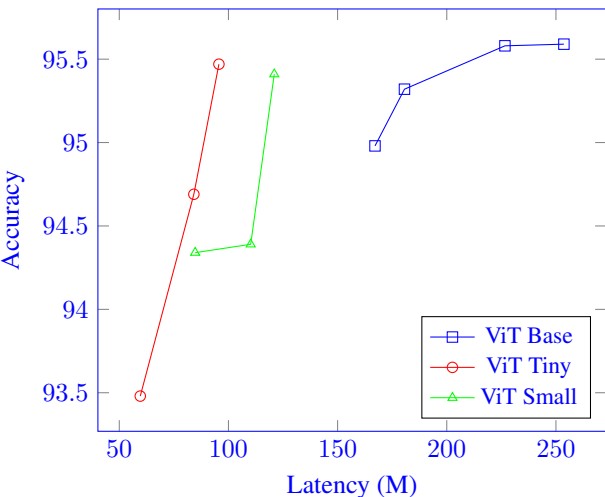

Figure 8: PriViT produces similar latency accuracy trade off for a variety of ViT models over CIFAR 10, highlighting it's scalability to large model sizes

**Analysis of performance degradation.** In this analysis, we aim to compare the performance of trained PriViT models with their finetuned versions. Our analysis is based on the class-level accuracy metric from the Tiny ImageNet dataset, which consists of 200 classes. We focus on three specific parameters to understand the performance degradation:

*Maximum Difference in Accuracy:* We assess the greatest disparity in accuracy across all 200 classes between the PriViT and finetuned models.

*Overall Accuracy Difference:* We compute the average accuracy difference between the finetuned and the PriViT models across all 200 classes.

*Variance in Accuracy Difference:* We analyze the consistency of the differences in accuracy across the 200 classes by calculating the variance.

Table 10: Performance degradation of PriViT models compared to finetuned model on tinyimagenet.

| Accuracy | Latency (M) | Max Difference | Mean Difference | Variance ($10 \times 10^{-3}$) |
|---|---|---|---|---|
| 69.8 | 151.75 | 30.00% | 1.85% | 6.8 |
| 66.98 | 128.23 | 34.00% | 4.68% | 6.9 |
| 64.46 | 110.60 | 34.00% | 7.21% | 8 |
| 60.53 | 93.72 | 40.00% | 11.13% | 9 |
| 59.55 | 86.51 | 34.00% | 12.12% | 9.4 |
| 59.58 | 84.78 | 36.00% | 12.08% | 9.8 |
| 59.04 | 84.13 | 36.00% | 12.63% | 9.9 |
| 58.74 | 69.42 | 40.00% | 12.92% | 9.4 |
| 58.2 | 67.43 | 36.00% | 13.48% | 8.7 |

Table 10 highlights that average accuracy degradation is anywhere between 1-13% for different non-linearity budgets but certain classes seem to be more adversely affected even in low budgets as the max class level difference in accuracy is consistent around 30%.

**Hyperparameter Tuning.** Following (Hassani et al., 2021) we use CutMix (Yun et al., 2019), Mixup (Zhang et al., 2017), Randaugment (Cubuk et al., 2020), and Random Erasing (Zhong et al., 2020) as data augmentation strategy. We probed multiple hyperparameter strategies for the joint optimization phase of PriViT to ensure consistent good performance over multiple configurations of non-linearity budgets of softmax and GELUs. Specifically we describe these strategies as follows:

**Late-Binarized Epoch (Strategy 1)**: This strategy involved 10 post-linearization training epochs. The binarization of auxiliary parameters, $s$ and $c$, occurred late in the process, specifically after the linearization was complete. The penalty increment condition for this method was checked when the reduction in the softmax and GELU coefficients per epoch was less than 200 and 2, respectively. Both masks began with identical penalties, signifying an 'equal' starting penalty.

**Late-Binarized Incremental (Strategy 2)**: This strategy also encompassed 10 training epochs with late binarization. Here, the penalty increment condition was activated with an increase in the softmax and GELU coefficients per epoch. The starting penalty for both masks was 'equal'.

**Late-Binarized Divergent Penalty (Strategy 3)**: Much like Strategy 2, this involved 10 epochs with late binarization and an increment condition based on softmax and GELU coefficient rises. However, the initial penalty was set to 'unequal', making the softmax penalty 20 times higher than the GELU penalty.

**Early-Binarized Incremental (Strategy 4)**: This strategy shared several similarities with Strategy 2, including 10 training epochs and an increment condition based on coefficient increases. The difference, however, lay in its early binarization, occurring during the freezing of the auxiliary parameters. The starting penalty was kept 'equal' for both masks.

**Prolonged Early-Binarized Epoch (Strategy 5)**: Spanning 50 post-linearization training epochs, this strategy adopted an early binarization approach. The penalty increment condition was activated when the reduction in softmax and GELU coefficients per epoch was under 200 and 2, respectively. The masks were initialized with 'equal' penalties.

Each of these strategies offered unique configurations in terms of epoch durations, binarization timings, increment conditions, and starting penalties, enabling a comprehensive assessment of the PriViT algorithm's performance under various conditions.

We test the different finetuning strategies described here by taylorizing PriViT for different softmax and GELU budgets and compare the test accuracy of the resulting model over CIFAR100. Table 11 highlights the comparative performance of all the strategies that we described. Strategy 5 seems to be performing best over different configuration of nonlinearity budget which is important as we would want to find the best model peformance for a particular non-linearity budget.

**Grid search of softmax and GELU configuration.** In order to elucidate the nuanced trade-off between softmax and GeLU operations, we executed a systematic grid search across an extensive parameter space encompassing varied softmax and GeLU configurations. Upon analysis of models

Table 11: We test the different finetuning strategies described in A. We run PriViT for different softmax and GELU budgets and compare the test accuracy of the resulting model over CIFAR100. We observe that strategy 5 works the best across a wide range of target softmax and GELU budgets.

| # Softmax (K) | # Gelu (K) | Strategy 1 (Acc. %) | Strategy 2 (Acc. %) | Strategy 3 (Acc. %) | Strategy 4 (Acc. %) | Strategy 5 (Acc. %) |
|---|---|---|---|---|---|---|
| 10 | 5 | 77.68 | 76.74 | - | 77.82 | **78.83** |
| 5 | 5 | 76.27 | 75.99 | 75.72 | - | **77.63** |
| 5 | 1 | 76.73 | 75.21 | 76.24 | - | **77.08** |
| 2 | 10 | 76.04 | 75.23 | - | 74.65 | **76.35** |
| 2 | 1 | 75.92 | 74.84 | 76.45 | - | **76.97** |
| 1 | 5 | 76.12 | 74.99 | 76.32 | - | **76.96** |

exhibiting iso-latencies, as demarcated by the red lines in figure 9, it became evident that the trade-off dynamics are non-trivial. Specifically, configurations with augmented softmax values occasionally demonstrated enhanced performance metrics, whereas in other scenarios, models optimized with increased GeLU counts exhibited superior benchmark results.

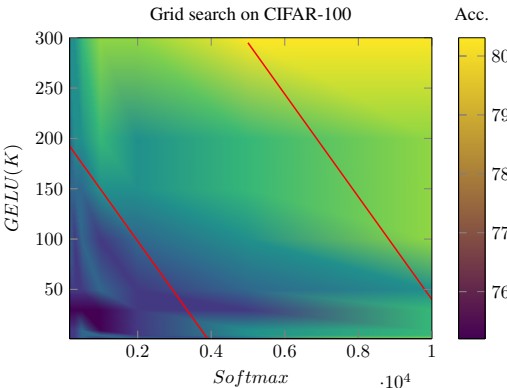

Figure 9: The PriViT algorithm produces a Pareto surface mapping the tradeoff between GeLU and softmax budgets over cifar 100.

**Taylorizing only one type of non-linearity.** The PriViT algorithm's standout capability is its simultaneous linearization of GELU and softmax operations, enabling a myriad of model configurations. In our focused experiment, we exclusively linearized GELU operations and anchored the auxiliary softmax parameter $S$, binarizing it to activate only the SoftmaxAttention mechanism. Despite extensive GELU substitutions, as reported in 10 the PriViT model displayed notable resilience on CIFAR10 and CIFAR100 datasets, with only slight performance drops, underscoring its robustness in varied setups.

**Effect of using pre-trained checkpoints.** To further investigate why using pretrained checkpoint is improving performance, we report the non-linear distributions searched by PriViT and compare it with PriViT without pretrain for the nonlinearity budget of 315k and 320k respectively. We observe from our findings in figures 11,12 that the distribution found by the two methods differs across each layer. This supports our theory as to how PriViT operates under a strategic 'top-down' paradigm. Starting with a fine-tuned model, it has the advantage of an architecture that has not just discerned overarching generalization patterns but has also selectively pruned irrelevant information, streamlining its focus for a specific downstream task. This reduction of redundancy, undertaken from a vantage point of a pre-existing knowledge base, gives PriViT an edge.

## B    SUPPLEMENTARY GRAPHICS

The following figure shows a graphical representation of the switching operation.

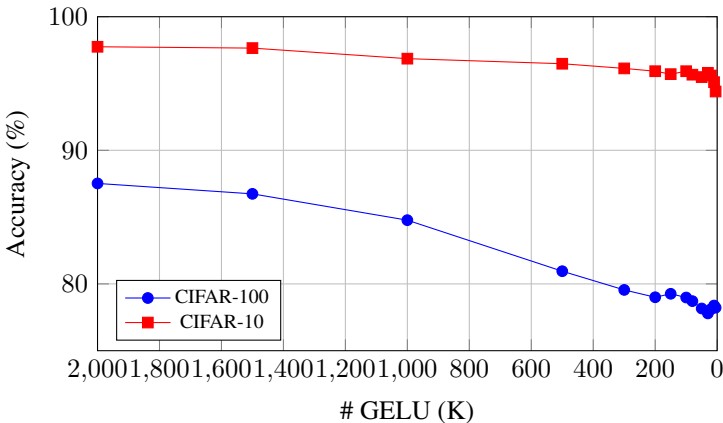

Figure 10: PriViT's ability to linearize GeLU operations visualized through performance on CIFAR datasets. As GELU operations decrease, CIFAR-100 and CIFAR-10 accuracies are affected, showcasing the trade-off between operation count and accuracy.

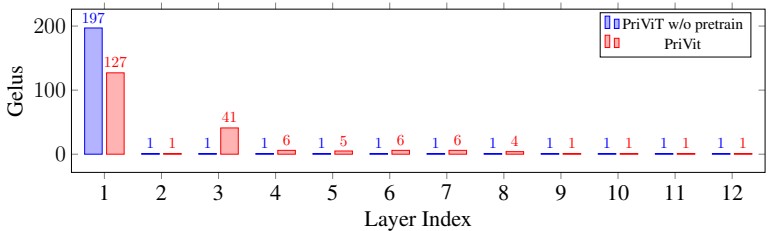

Figure 11: We compare the distribution of 208 GELU and 200 GELU operations distributed by PriViT w/o pretrain and PriViT respectively over tiny imagenet dataset.

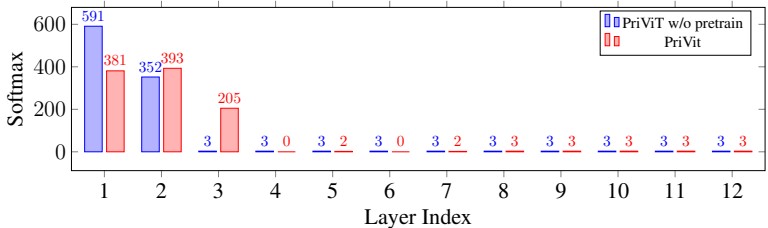

Figure 12: We compare the distribution of 973 softmax operations and 998 softmax operations operations distributed by PriViT w/o pretrain and PriViT respectively over tiny imagenet dataset.

**Search granularity.** An important characteristic of PriViT is it's flexibility to search over different granularity of non-linearities. GELU is a pointwise functions, thus PriViT can search either at embedding level or at a token level. On the other hand, softmax is a token level operation, thus it cannot be broken into a finer search space. Note that softmax operations can be extended to search over the head space or layer space, and similarly GELU can be searched over the layer space. Figure 14 illustrates the search granularity over token and embedding space.

**Parametric mask.** Figure 13 is an illustration of the working mechanism of the parametric mask introduced in PriViT. When the parameters are binarized it selects one of the two candidate function in the attention mechanism, and the gelu activation.

**PriViT overview.** Figure 15 provides an illustration of the complete PriViT algorithm, there are three distinct phases namely Finetuning the teacher, joint optimization of network and parametric masks, and final finetuning.

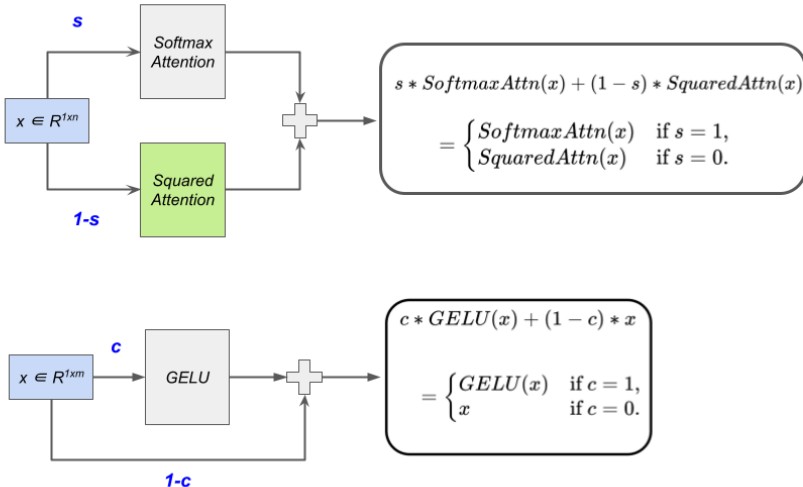

Figure 13: Parameterized Gelu and Self-Attention operations. **Top**: Tokens undergo softmax and squared attention in training. Post-training, parameter $S$ is frozen and binarized, selecting only one operation. **Bottom**: Embeddings pass through GeLU and Identity during training. Afterwards, parameter $C$ is frozen and binarized, choosing a single operation.

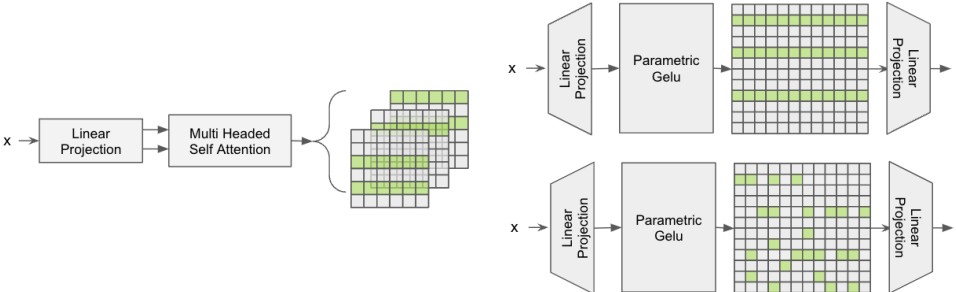

Figure 14: **Left**: The green blocks are SQUAREDATTENTION, and the grey blocks are Softmax Attention. For parametric attention, tokens emerge from a blend of softmax and square attention (refer to fig 13). Post-training, auxiliary variable $S$ is set to 0 or 1, resulting in $2^{N \times H}$ potential combinations per encoder block. **Right**: The green blocks are Identity function, and the grey blocks are GELU activation. Embeddings combine GELU and identity operations during training, as seen in fig 13. After training, parameter $C$ is frozen and binarized. This yields potential combinations of either $2^{H \times N}$ or $2^{N \times H \times m}$ for each ViT encoder block. Note that GELU being a pointwise function, we possess the flexibility to expand our search space either to tokens or directly to individual embeddings.

## C  PRIVIT ALGORITHM

We provide detailed pseudocode to implement the PriViT algorithm in Algorithm 1.

## D  ROLE OF DELPHI IN OUR FRAMEWORK

Delphi Mishra et al. (2020a) encompasses two primary elements: a secure communication protocol and a NAS technique, specifically an evolutionary method. The secure communication aspect focuses on cryptographic protocols for neural networks within a multi-party computation setting. The NAS technique, known as Delphi's planner, is designed primarily to eliminate ReLU operations from neural architectures.

---

**Algorithm 1** PRIVIT: Privacy Friendly ViT

---

1: **Inputs:** $f_{\mathbf{W}}$: pre-trained network, $\lambda_s$: Lasso coefficient for Softmax mask, $\lambda_g$: Lasso coefficient for GeLU mask, $\kappa$: scheduling factor, $G$: GeLU budget, $S$: Softmax budget, $\epsilon$: threshold.
2: Set $\mathbf{C} = 1$: same dimensions to all GeLU mask.
3: Set $\mathbf{S} = 1$: same dimensions to all Attention Heads.
4: Set $C_{\text{budget}} = False$: GeLU budget flag.
5: Set $S_{\text{budget}} = False$: Softmax budget flag.
6: $\overline{\mathbf{W}} \leftarrow (\mathbf{W}, \mathbf{C}, \mathbf{S})$
7: Lowest GeLU Count $\leftarrow \|\mathbb{1}(\mathbf{C} > \epsilon)\|_0$
8: Lowest Softmax Count $\leftarrow \|\mathbb{1}(\mathbf{S} > \epsilon)\|_0$
9: **while** GeLU Count $> G$ or Softmax Count $> S$ **do**
10:     Update $\overline{\mathbf{W}}$ via ADAM for one epoch.
11:     GeLU Count $\leftarrow \|\mathbb{1}(\mathbf{C} > \epsilon)\|_0$
12:     Softmax Count $\leftarrow \|\mathbb{1}(\mathbf{S} > \epsilon)\|_0$
13:     **if** Lowest GeLU Count - GeLU Count $< 2$ **then**
14:         $\lambda_g \leftarrow \kappa \cdot \lambda_g$.
15:     **end if**
16:     **if** Lowest Softmax Count - Softmax Count $< 200$ and $S_{\text{budget}} = False$ **then**
17:         $\lambda_s \leftarrow \kappa \cdot \lambda_s$.
18:     **end if**
19:     **if** Lowest GeLU Count $>$ GeLU Count **then**
20:         Lowest GeLU Count $\leftarrow$ GeLU Count
21:     **end if**
22:     **if** Lowest Softmax Count $>$ Softmax Count **then**
23:         Lowest Softmax Count $\leftarrow$ Softmax Count
24:     **end if**
25:     **if** GeLU count $<=$ G and $C_{\text{budget}} = False$ **then**
26:         $\mathbf{C} \leftarrow \mathbb{1}(\mathbf{C} > \epsilon)$
27:         $C_{\text{budget}} = True$
28:         $\overline{\mathbf{W}} \leftarrow (\mathbf{W}, \mathbf{S})$
29:     **end if**
30:     **if** Softmax count $<=$ S and $S_{\text{budget}} = False$ **then**
31:         $\mathbf{S} \leftarrow \mathbb{1}(\mathbf{S} > \epsilon)$
32:         $S_{\text{budget}} = True$
33:         $\overline{\mathbf{W}} \leftarrow (\mathbf{W}, \mathbf{C})$
34:     **end if**
35: **end while**

---

In our paper, we utilize Delphi's secure communication protocol for private inference. Furthermore, we took inspiration from Delphi's planner to design an effective method to reduce non-linear operations for attention-based architectures, particularly in Vision Transformers (ViTs). This enhancement was necessary because the original Delphi's planner is not optimally equipped for reducing non-linearities like softmax, gelu, and layernorms in ViT architectures.

Figure 16 provides an estimate of Delphi's planner lower bound on latency on ViT-Base with 80M parameters. To estimate this lower bound, we run PriVit on ViT-Base but only replace GELU operations with identity operations, till no Gelus were left to linearize, i.e. the left most point in the cyan curve indicates a network with no Gelus. All softmax and layernorms are left intact, Delphi's planner lower bound on latency is defined by the red vertical line which represents the latency of a model with no GELUs, and all softmax/layernorm intact. We compare this lower bound with the latency accuracy curve produced by PriViT on ViT-Base that Taylorize both softmax and GELUs.

## E    ATTENTION VARIANTS

Here we describe formally the different attention variant we ablated. Uniform form attention is basically described by the following equation

$$\text{UNIFORMATTN}(\mathbf{X}) = \frac{(1)}{N} \mathbf{X} \mathbf{W}_v, \tag{8}$$

Where N is the number of tokens, so for each token the attention weights are equal hence the name UniformAttention.

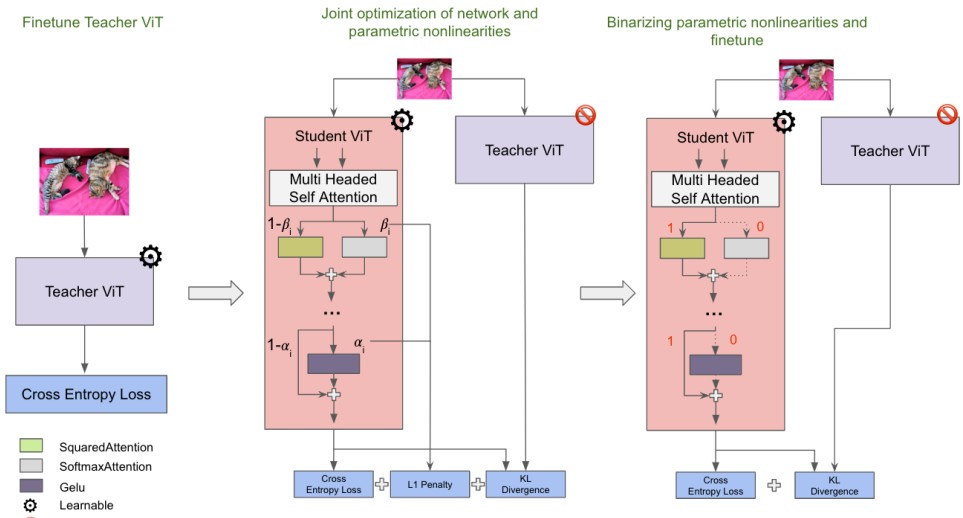

Figure 15: **Left**: Step 1 - Fine-tuning of a pretrained ViT over target dataset to produce the 'teacher ViT'. **Middle**: Step 2 - Duplicate teacher ViT, introduce parametric GELUs and attention mask to form 'student ViT'. Train using cross-entropy loss, KL divergence, and L1 penalty to gradually find a sparse mask. Binarize the mask post desired non-linearity budget. **Right**: Step 3 - With a frozen, binarized mask, further fine-tune the student model using cross-entropy loss and KL divergence with the teacher.

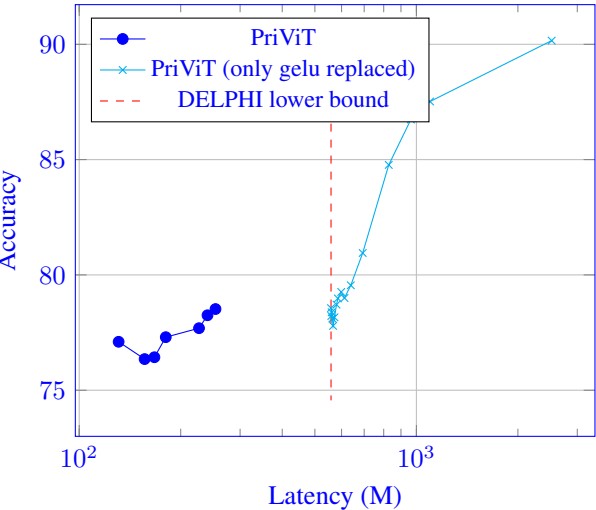

Figure 16: Estimation of Delphi's latency lower bound compared to PriViT. We use ViT-Base as our architecture and CIFAR 100 as our dataset.

ScaleAttn is the softmax candidate used in the work Zeng et al. (2022) which is essentially described as

$$\text{SCALEATTN}(\mathbf{X}) = \frac{\left(\mathbf{X}\mathbf{W}_q\mathbf{W}_k^T\mathbf{X}^T\right)}{N}\mathbf{X}\mathbf{W}_v, \tag{9}$$

