| PriViT G | | MPCViT | | PriVit R | | MPCViT+ | |
|---|---|---|---|---|---|---|---|
| Acc | Latency (s) | Acc | Latency (s) | Acc | Latency (s) | Acc | Latency (s) |
| 96.31 | 21.13 | 94.3 | 10.39 | 94.45 | 18.74 | 93.3 | 5.57 |
| 95.31 | 19.27 | 94.2 | 9.85 | 92.45 | 17.83 | 93.9 | 6.38 |
| 95.58 | 15.99 | 94.1 | 9.39 | 92.36 | 13.26 | 94.2 | 7.39 |
| 95.14 | 14.43 | 93.6 | 8.96 | 92.39 | 13.01 | 94.3 | 6.83 |
| 94.52 | 14.37 | | | 91.08 | 10.24 | | |
| 94.44 | 11.6 | | | | | | |

Table 6: Benchmarking PriViT and MPCViT over CIFAR 10 dataset on SEMI2k protocol.

## A   SUPPLEMENTARY RESULTS

**Additional PI results** Following Zeng et al. (2022) we benchmark our method over SEMI2k using secretflow framework, the client and server are 64GB RAM, Intel(R) Xeon(R) Platinum 8268 CPU @ 2.90GHz. We run PI over LAN settings between two nodes of HPC cluster, hence there is a variation of the total inference latency from what is reported in Zeng et al. (2022), but to keep a consistent comparison, we benchmark both PriViT and MPCViT under our system settings. We report additional bench marking results on CIFAR 10 data in the table 6.

**Analysis of performance degradation.** In this analysis, we aim to compare the performance of trained PriViT models with their finetuned versions. Our analysis is based on the class-level accuracy metric from the Tiny ImageNet dataset, which consists of 200 classes. We focus on three specific parameters to understand the performance degradation:

*Maximum Difference in Accuracy:* We assess the greatest disparity in accuracy across all 200 classes between the PriViT and finetuned models.

*Overall Accuracy Difference:* We compute the average accuracy difference between the finetuned and the PriViT models across all 200 classes.

*Variance in Accuracy Difference:* We analyze the consistency of the differences in accuracy across the 200 classes by calculating the variance.

Table 7 highlights that average accuracy degradation is anywhere between 1-13% for different non-linearity budgets but certain classes seem to be more adversely affected even in low budgets as the max class level difference in accuracy is consistent around 30%.

Table 7: Performance degradation of PriViT models compared to finetuned model on tinyimagenet.

| Accuracy | Latency (M) | Max Difference | Mean Difference | Variance ($10 \times 10^{-3}$) |
|---|---|---|---|---|
| 69.8 | 151.75 | 30.00% | 1.85% | 6.8 |
| 66.98 | 128.23 | 34.00% | 4.68% | 6.9 |
| 64.46 | 110.60 | 34.00% | 7.21% | 8 |
| 60.53 | 93.72 | 40.00% | 11.13% | 9 |
| 59.55 | 86.51 | 34.00% | 12.12% | 9.4 |
| 59.58 | 84.78 | 36.00% | 12.08% | 9.8 |
| 59.04 | 84.13 | 36.00% | 12.63% | 9.9 |
| 58.74 | 69.42 | 40.00% | 12.92% | 9.4 |
| 58.2 | 67.43 | 36.00% | 13.48% | 8.7 |

**Fine-grained versus layer-wise Taylorization** PriVit employs a unique approach where it selectively Taylorizes softmax and GELU operations in models. To probe the effectiveness of this method, we contrasted it with an alternative PriViT approach that Taylorizes a ViT model progressively, layer by layer.

As illustrated in Table 8, our observations underscored the superiority of selective Taylorization.

Table 8: Performance comparison of PriViT versus layerwise linearization of GeLU in a ViT model with 200k GeLUs. Twelve models were generated by sequentially replacing up to 12 GeLU layers with Identity. PriViT was also evaluated with varying GeLU budgets below 200k.

| Layerwise GELU linearizing | | Pri-ViT | |
| --- | --- | --- | --- |
| Gelus (K) | Accuracy | Gelus (K) | Accuracy |
| 197 | 96.07 | 200 | 95.59 |
| 196 | 96.07 | 150 | 95.34 |
| 193 | 95.91 | 100 | 95.58 |
| 190 | 95.35 | 50 | 94.98 |
| 187 | 94.28 | 10 | 94.24 |
| 184 | 93.75 | 1 | 93.96 |
| 181 | 93.33 | | |
| 178 | 92.99 | | |
| 174 | 93.04 | | |
| 171 | 92.88 | | |
| 164 | 92.06 | | |
| 123 | 82.48 | | |
| 0 | 56.64 | | |

This superiority was especially pronounced under constrained non-linearity budgets.

Delving deeper, our experiment commenced with a foundational ViT model populated with 200k GeLUs, while the remaining operations were Identity-based. From this foundation, we crafted a series of models, each with an increasing number of GeLU layers swapped for Identity, creating a spectrum from 1 to 12 GeLU replacements. Post-finetuning, the performance metrics of these models were recorded. In parallel, we evaluated PriViT under a gamut of GeLU budgets, all set below the 200k threshold, thereby exploring its capability for dynamic GeLU retention.

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

## D  LATENCY BENCHMARKS

We conduct thorough benchmarking by creating GC circuits for the non-linearity functions found in ViT, and also benchmark specific functions used in MPCViT so as to enable us to compare the two methods under the same protocol DELPHI. In order to compare the different cost of non-linearity we bring them down to a common benchmark of ReluOps, where 1 ReluOp is the cost incurred for performing a GC evaluation of ReLU of one scalar value. Figure 14 shows how we count the non-linearity cost of softmax. The front and end consider Secret Sharing similar to Circa Ghodsi et al. (2021). Since the GC cost of each operation is known, we add them up as the final cost of softmax.

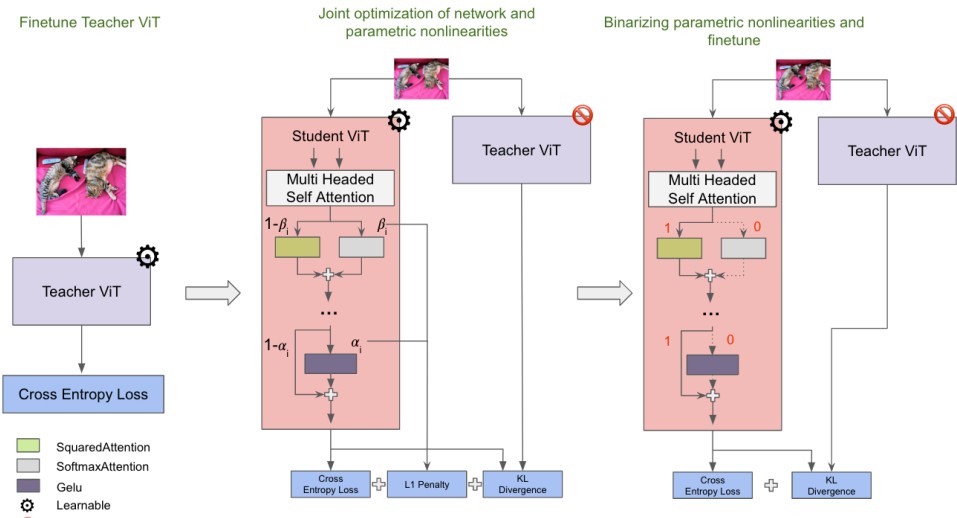

Figure 13: **Left**: Step 1 - Fine-tuning of a pretrained ViT over target dataset to produce the 'teacher ViT'. **Middle**: Step 2 - Duplicate teacher ViT, introduce parametric GELUs and attention mask to form 'student ViT'. Train using cross-entropy loss, KL divergence, and L1 penalty to gradually find a sparse mask. Binarize the mask post desired non-linearity budget. **Right**: Step 3 - With a frozen, binarized mask, further fine-tune the student model using cross-entropy loss and KL divergence with the teacher.

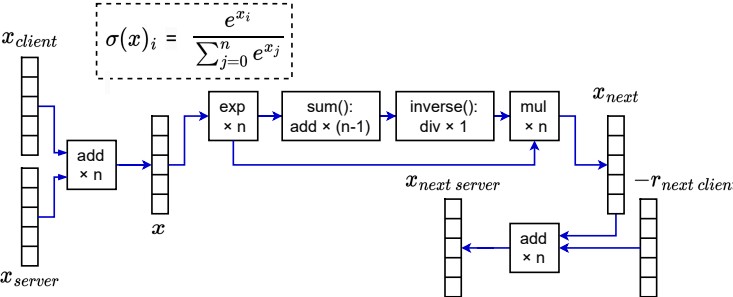

Figure 14: *Detailed steps of benchmarking the non-linearity cost for softmax. Denominator is calculated once and reused for all indices of the vector.*

## E  ATTENTION VARIANTS

Here we describe formally the different attention variant we ablated. Uniform form attention is basically described by the following equation

$$\text{UNIFORMATTN}(\mathbf{X}) = \frac{(1)}{N}\mathbf{W}_v\mathbf{X}, \tag{7}$$

Where N is the number of tokens, so for each token the attention is equal hence the name UniformAttention.

ScaleAttn is the softmax candidate used in the work Zeng et al. (2022) which is essentially described as

$$\text{SCALEATTN}(\mathbf{X}) = \frac{(\mathbf{X}\mathbf{W}_q\mathbf{W}_k\mathbf{X})}{N}\mathbf{W}_v\mathbf{X}, \tag{8}$$