# OpenReview forum: "PriViT: Vision Transformers for Fast Private Inference"
_ICLR.cc/2024/Conference — Submitted to ICLR 2024_

### Official Review · Reviewer_pfyb · 2023-10-29

**Soundness:** 3 good
**Presentation:** 2 fair
**Contribution:** 2 fair
**Rating:** 5
**Confidence:** 3

**Summary:**

The paper introduces "PriViT," an algorithm designed to make Vision Transformers (ViTs) suitable for private inference using secure multi-party computation (MPC) protocols. Traditional ViTs are not ideal for MPC due to their complex non-polynomial operations. PriViT aims to reduce the nonlinearities in ViTs and preserve their prediction accuracy. It demonstrates a balance in latency and accuracy superior to existing methods, as confirmed through experiments on image classification tasks.

**Strengths:**

The paper is easy to follow. The paper tackles the significant challenge of making Vision Transformers (ViTs) compatible with private inference using secure multi-party computation (MPC) protocols.

**Weaknesses:**

A predominant weakness of the paper is its limited novelty, as it appears to draw heavily on core concepts previously established in SNL.

**Questions:**

1.	While the paper introduces an interesting approach, its novelty appears constrained. The core concept of utilizing binary gates to manage the number of nonlinear operations through $\ell_0$ sparsity has been proposed introduced in SNL (Cho et al., 2022b). The proposed method seems like an adaptation of SNL for ViTs. This adaptation is intriguing, but a deeper exploration into its unique contributions compared to SNL would provide more clarity on its novelty.

2.	The paper's organization raises some concerns. While critical figures, and experimental results are put in the supplementary material, the main body of the paper exhibits noticeable empty spaces. It would enhance the paper's comprehensibility and impact if these essential elements were incorporated directly into the main content, making the narrative more coherent for readers without necessitating frequent references to supplementary sections.

3.	In Figure 1, the computational cost is quantified using the metric of # AND gates. However, the methodology or criteria for determining the number of AND gates associated with a nonlinear operation remains ambiguous in the paper. Clarifying this calculation or providing a concise explanation within the main content would enhance the reader's understanding.

4.	The experimental validation is primarily focused on small-scale datasets, which could limit the generalizability of the findings. For a comprehensive evaluation, it would be beneficial for the authors to include results from large-scale datasets like ImageNet. Such experiments would provide a more holistic view of the method's efficacy and scalability.

5.	Based on Figure 2, both MPCViT and MPCViT+ seem to outperform the proposed method in achieving a superior Pareto balance between accuracy and latency. This observation raises the question: what distinctive advantages or contributions does the proposed method offer? Is it perhaps more streamlined in terms of training efficiency or some other metric? A clearer delineation of the method's unique strengths would be beneficial for readers.

6.	Observations from Table 4 present some anomalies. Specifically, for PriViT-R, there is a counterintuitive trend where increased latency corresponds to a decline in performance. This is perplexing as one would typically expect a trade-off where longer computation times would yield better results. An explanation or insight into this apparent contradiction would enhance the clarity of the findings.

7.	The clarity of notational conventions could be enhanced. For instance, the terms PriViT-R and PriViT-G appear in the text without explicit definitions or context. Providing a clear explanation or description of what these notations signify would aid in a more comprehensive understanding of the content for readers.

---

> ### Author Response · Authors · 2023-11-22
> **Response**
>
> We thank the reviewer for the constructive feedback. We are glad that the reviewer thinks the paper is easy to understand and tackles a significant challenge (enabling secure MPC for ViTs).
>
> **PriViT's novelty compared to SNL**: It is important to highlight that the SNL algorithm is equipped to linearize only ReLU activations, and more generally is applicable to networks that only exhibit a single type of linearity. The algorithmic innovation in PriViT is to design a method that can jointly linearize two or more types of non-linearities. Additionally, we equip our method to Taylorize the attention mechanism with an appropriate candidate function to make it suitable not only for ViTs, but potentially any future architecture based on attention mechanisms. Finally we propose a GELU replacement strategy to further reduce the latency of ViTs in low latency budget settings.
>
> **Paper organization**: We have revised our paper to incorporate your suggestions, we hope the new version is easy to follow for readers. All changes are highlighted in blue, and many figures and tables are pulled from the Appendix into the main paper.
>
> **Calculation of AND gates**: The method to determine the number of AND gates is by first breaking down a complex nonlinear function into several basic operations. These operations are already synthesized [TinyGarble] and included in the EMP-Tool library (a commonly-used GC library). Thus we simply add all the AND gates of these basic operations to arrive at the total number of AND gates. We moved relevant figures to the main paper from Appendix to improve readability. Section 2, Figure 2 mentions this explanation in the main paper.
>
> **PriViT on ImageNet**: The suggestion to test PriViT on datasets like ImageNet that have resolution higher than 64 by 64 is essential to highlight the universality of this method. We would like to refer to Section 4.1 (paragraph titled Architecture and data set) where we mention that all images are up sampled to 224 by 224 before being passed in as input to the ViT. Hence all the results presented in the paper are based on the resolution of 224 by 224. To our best knowledge, SENet has not provided publicly available code to train their models, which makes quite difficult to reproduce their method for ViTs.
>
> **PriViT's advantage compared to MPCViT**: Based on your suggestions we'd like to present some additional data to highlight the efficient training methodology of PriViT compared to MPCViT. Since training codes are not public for MPCViT, we cannot compare GPU memory usage, so we only compare the number of training epochs as it is reported in their paper. MPCViT undergoes a NAS stage where an architecture is searched for 300 epochs, followed by 600 and 300 epochs of training the searched model on CIFAR 100 and TinyImagenet respectively, whereas PriViT simply trains a ViT in one go. Table 5 in the main paper highlights that for an isoaccuracy of ~64% PriViT requires ~50% less training epochs compared to MPCViT on TinyImagenet. On CIFAR 100 for an isolatency of ~75 M PriViT requires ~50% of the training epochs of MPCViT on TinyImagenet.
>
> Thus PriViT not only achieves a superior performance on latency, but it does so in a way that is highly efficient, and scalable to larger datasets. This makes PriViT an ideal choice for private inference when a business has personalized models for their individual users, and would like to make them PI friendly.
>
> **PriViT-R anomalous behavior**: The unexpected performance of PriViT in Table 4 can be attributed to how PriViT R models are derived. By replacing GELUs in PriViT G models (developed as per our algorithm) with ReLUs, we create PriViT R models. Although individual ReLU and GELU are similar, repeated replacements amplify errors, impacting the model's performance, especially at high GELU budgets. This explains why models with higher latency (82M) underperform compared to those with lower latency. We have revised our paper to recommend PriViT R's use only at low GELU budgets, where it effectively reduces latency without compromising performance.
>
> **Notation clarity**: We have updated our main paper to give more clarity on our notations. All updates are in blue.

---

### Official Review · Reviewer_BazG · 2023-10-31

**Soundness:** 3 good
**Presentation:** 2 fair
**Contribution:** 3 good
**Rating:** 6
**Confidence:** 3

**Summary:**

This paper introduces PriViT as a solution for implementing Vision Transformers~(ViTs) in private inference (PI) applications. To achieve this, PriViT tailors the nonlinear operations (e.g., GELU and row-wise softmax) of a pre-trained Transformer in order to make them compatible with secure multi-party computation (MPC) protocols. This adaptation involves the introduction of learnable switching variables to gradually replace the original nonlinear operations. The experimental results on CIFAR-10/100 and TinyImageNet demonstrate a significant improvement in both speed and performance compared to prior approaches.

**Strengths:**

* The paper is well-motivated as the deployment of ViTs in private scenarios is becoming increasingly important and current approaches are not tailored for Transformer architecture.
* The proposed method is quite simple yet effective comparing with SOTA approaches.

**Weaknesses:**

[Major]

1. **Experiments:** The authors have conducted sufficient comparative experiments conducted on 3 datasets (CIFAR-10/100 and TinyImageNet). However, the image resolutions are no more than $64\times 64$, which is rather small compared with commonly-used datasets like ImageNet-1k and Caltech-101/256. It will be interesting to see ImageNet results and compare with SENet if possible.
2. **Experiments:** The authors' exclusive use of ViT-Tiny for comparison is insufficient to establish the method's universality. It is strongly recommended that the authors broaden their evaluation to include a variety of ViT architectures, such as DeiT, and models with diverse parameters, including ViT-Small and ViT-Base. Such an expanded assessment would be of significant interest to the community, particularly in light of the flourishing development of large Transformer-based vision models.
3. **Experiments:** In Table 3, the authors have presented PriViT and MPCViT in distinct hyper-parameter configurations, giving rise to concerns regarding the fairness of the comparisons.
4. **Experiments:** Furthermore, I strongly encourage the authors to furnish additional results to showcase the superiority of their proposed method. This could encompass metrics like GPU memory usage and training time in comparison to other existing methods.

[Minor]
1. This article is poorly written, with issues including imprecise mathematical formulas, non-standard captions, typographical errors, and punctuation. Here, I have listed some errors and hope that the authors will carefully revise and proofread their manuscript and the appendices.
    * Eq (1): $o=\frac{\mathrm{Softmax}(XW_qW_kX)}{\sqrt{d}}W_vX$ -> $o=\frac{\mathrm{Softmax}(XW_qW_k^\top X^\top)}{\sqrt{d}}XW_v$. Eq (4) has the same problem.
    * Page 5, last paragraph: Once the model satisfies the required budgets,,we... -> Once the model satisfies the required budgets, we...
    * Page 6, first paragraph: Architecture and data set. -> Architecture and dataset.
    * Page 6, first paragraph: ViT Tiny -> ViT-Tiny
    * Page 6, first paragraph: 224x224 -> $224\times 224$
    * The caption of tables should be on the top of the table: Table 3 and 5.
    * Page 7: Pareto analysis of PriViT over Tiny Imagenet, and Cifar10/100 -> Pareto analysis of PriViT over Tiny Imagenet, and Cifar10/100.
    * Page 7: table 4 -> Table 4, Fig 2 -> Figure 2, fig 13 -> Figure 13, figure 3 -> Figure 3, Fig. 13 -> Figure 13.
    * Page 8: The latency is calculated as per 4.1 -> The latency is calculated as per Section 4.1.
2. Missing dataset details in Table 1.
3. The authors do not discuss the limitations of their method.
4. The authors do not provide codes for reproducibility check.

**Questions:**

My questions are listed in the "Weaknesses" section. I am looking forward to the authors' relply.

---

> ### Author Response · Authors · 2023-11-22
> **Response**
>
> We thank the reviewer for the constructive feedback. We are glad that you think the paper solves a well-motivated and important problem, and that our method is simple yet effective.
>
> **PriViT on Imagenet**: We support the suggestion to test PriViT on datasets such as Imagenet and Caltech-101/256, that have a native resolution higher than 64x64. We would like to point out that  in Section 4.1 (paragraph titled Architecture and data set) where we mention that all images are upsampled to 224x224 before being passed in as input to the ViT, so our PriViT algorithm and implementation natively works for ImageNet resolutions without any changes. The sole reason for focusing on CIFAR-10/100 and TinyImageNet in our experiments is that almost every previous approach on private inference uses these smaller (and lower-resolution) datasets as benchmarks.
>
> **PriViT on different architectures**: Fig 8 provides a comparison of PriViT over different model variants. We see that PriViT is able to produce a similar latency-accuracy tradeoff across different model sizes, emphasizing its generalizability.
>
> **PriViT's comparisons with MPCViT**: We acknowledge the concerns about the fairness of comparisons in Table 3 due to differing hyper-parameter configurations. To re-iterate, PriViT is implemented at 224x224 resolution, which aligns with more challenging “real-world” benchmarks such as ImageNet. This is in comparison with MPCViT, which only supports lower resolutions. This crucial distinction underscores PriViT's broader applicability and superior adaptability. Despite the comparison not being apples-to-apples in hyper-parameters, it demonstrates PriViT's robust performance across more realistic scenarios, taking a step towards private inference for practical transformer architectures.
>
> **PriViT's advantage compared to MPCViT**: Based on your suggestions we'd like to present some additional data to highlight the efficient training methodology of PriViT compared to MPCViT. Since training codes for MPCViT have not been made public, we can only compare the number of training epochs as reported in their paper.
>
> MPCViT undergoes a NAS stage where an architecture is searched for 300 epochs, followed by 600 and 300 epochs of training the searched model on CIFAR 100 and TinyImagenet respectively, whereas PriViT simply trains a ViT in a single run. Table 5 highlights that for an isoaccuracy of ~64% PriViT requires ~50% less training epochs compared to MPCViT on TinyImagenet. On CIFAR 100 for an isolatency of ~75 M PriViT requires ~50% of the training epochs of MPCViT on TinyImagenet.
>
> Therefore, PriViT not only achieves a superior performance on latency, but it does so in a way that is efficient and scalable to larger/higher resolution datasets compared to MPCViT.
>
> **PriViT's paper organization**: We have revised our paper to incorporate your suggestions. We hope that the new version is easy to follow for readers. All changes are highlighted in blue, and many figures and tables are pulled from the Appendix into the main paper.

---

### Official Review · Reviewer_HTX8 · 2023-11-01

**Soundness:** 2 fair
**Presentation:** 3 good
**Contribution:** 2 fair
**Rating:** 5
**Confidence:** 2

**Summary:**

They propose a new vision transformer architecture which is a MPC-friendly ViT. The proposed architecture optimizes the attention part with the SquaredAttn and replace the GELU function to reduce the nonlinear operations in the model. The proposed method achieve state-of-the-art performance on several image classification benchmarks and speed up the model compared with current MPCViT.

**Strengths:**

1. The method analysis is clear and latency breakdown is helpful.
2. The experiments on serval image classification benchmarks are solid and comprehensive.
3. The proposed method is speed up than previous SOTA model and achieve competitive performance.

**Weaknesses:**

1. Need more detailed about the knowledge distillation part.
2. More discussion about non-linearity distribution.

**Questions:**

1. I am curious about the teacher model size and can it boost more performance if we use a larger teacher?
2. As you states that later layers have a larger number of linearized units, can we first try to linearized the later layers then try to linearized the earlier layers?

---

> ### Author Response · Authors · 2023-11-22
> **Response**
>
> We thank the reviewer for the constructive comments. In particular, we are glad that the reviewer appreciates the clarity of the method, the comprehensiveness of the benchmarks, and the competitive performance (with respect to SOTA) of our framework.
>
> **More discussions about knowledge distillation and nonlinearity distribution**: Thanks for this comment! Based on your suggestion, we have updated Section 3.3 to explain knowledge distillation (KD) with a mathematical equation. Additional discussion about KD can also be found in Section 4.1 (see paragraph titled Joint optimization of student ViT and parametric non linearities).
>
> As the main focus of this work is on accelerating private inference, we have conducted only a limited investigation about the distribution of non-linearities throughout the ViT. Please refer to Section 4.3, paragraph titled “Visualization of non-linearity distribution.”) Some more discussion about non-linear distribution is in Appendix A (please refer paragraph titled “Effect of using pre-trained checkpoints”), where we discuss how two ViTs with similar non-linearity budgets differ in their performance because of differences in their non-linearity distribution across encoder layers. Overall, we believe this is an interesting and important direction for future study.
>
> Based on your suggestion we are currently running experiments with different teacher models to see its effect on the performance. Due to time constraints, we could not finish our experiments with larger teacher models in this revision, although our preliminary results show promising improvements. We plan to report these findings in a future update to our work. The suggestion to linearize later layers first is a good one, although it is important to note that we do not know the distribution of nonlinearity beforehand (the PriViT algorithm automatically deduces this by optimizing for the switching patterns for all layers simultaneously).

---

### Official Review · Reviewer_y916 · 2023-11-04

**Soundness:** 2 fair
**Presentation:** 2 fair
**Contribution:** 2 fair
**Rating:** 5
**Confidence:** 3

**Summary:**

This paper proposed a new algorithm for constructing MPC-friendly vision transformers. Accuracy and latency performance were compared on several datasets with the previous result, MPCViT.

**Strengths:**

By adjusting GELU and softmax through training using a switched method, they found a different optimal method for each layer.

**Weaknesses:**

Compared to the prior technology, MPCViT, it shows better results in the TinyImagenet but worse latency in the CIFAR-100. In terms of accuracy, it has superior performance in any case. The paper said that DELPHI is focused as the subject of comparison. "In this paper, our focus is exclusively on the DELPHI protocol (Mishra et al., 2020a) for private inference. We choose DELPHI as a matter of convenience;" However, the actual results do not show any performance comparison with DELPHI.

**Questions:**

It is necessary to explain in what aspects DELPHI was considered.

**Details Of Ethics Concerns:**

I have no concern

---

> ### Author Response · Authors · 2023-11-22
> **Response**
>
> We thank the reviewer for the constructive feedback. The reviewer correctly notes that our PriViT approach (via a switching method) identifies “optimal” Taylorization strategies for ViT nonlinearities layer-by-layer.
>
> **Role of Delphi**: We appreciate the opportunity to clarify key components of the Delphi approach [1] as applied in our work. We added these points in the Appendix, Section D.1. New text and figures are highlighted in blue. Specifically, in the sentence that the reviewer quotes in their review, we are referring to the secure communication protocol adopted by Delphi, and not their architectural modifications. Let us explain further as follows.
>
> Delphi encompasses two primary elements: (a) a secure communication protocol, and (b) a NAS technique (specifically an evolutionary method) used to reduce ReLU nonlinearities. The secure communication part focuses on cryptographic protocols for neural networks within a two-party computation setting. The NAS technique, known as Delphi’s “planner”, is designed primarily to eliminate ReLU operations from neural architectures.
> Our PriViT framework utilizes Delphi’s secure communication protocol, which prescribes additive secret sharing for linear operations, and garbled circuits for nonlinearities. The point we were trying to make is that we adopted the Delphi protocol for no specific reason and purely out of convenience; one could imagine using other communication protocols instead (such as CryptFlow2 [2]). Our PriViT method would not materially change, but the latency numbers in all our benchmarks might be different. Overall, we expect that all experimental trends will persist.
>
> We are noting up front that Delphi is not well-suited for reducing non-linearities in ViT structures (softmax, GELUs, and layernorms.) However, for completeness, as per the reviewer’s request, we include now a brief comparison with the original Delphi’s “planner” stage as well.
>
> Figure 16 provides an estimate of Delphi’s planner lower bound on latency on ViT-Base with
> 80M parameters. To estimate this lower bound, we run PriVit on ViT-Base but only replace GELU operations with identity operations, till no Gelus were left to linearize, i.e. the left most point in the cyan curve indicates a network with no Gelus. All softmax and layernorms are left intact. Delphi’s planner lower bound on latency is defined by the red vertical line which represents the latency of a model with no GELUs, and all softmax operations and layer norms intact. We compare this lower bound with the latency accuracy curve produced by PriViT on ViT-Base that Taylorize both softmax and GELUs. Based on this observation, we see that PriViT achieves a significantly lower bound on latency compared to Delphi.
>
> [1] Mishra et al, ‘Delphi: A Cryptographic Inference Service for Neural Networks’, 2020.
> [2] Rathee et al, ‘CryptFlow2: Practical 2-Party Secure Inference’, 2020.

---

### Author Response · Authors · 2023-11-22
**Summary of responses**

We thank the reviewers for their insightful feedback. We are gratified by the positive reception of our proposed method, particularly its motivation, simplicity, and effectiveness in surpassing the performance of previous state-of-the-art in private inference for vision transformers. Their appreciation for our clear analysis, and our method's ability to address significant challenges in deploying vision transformers for private scenarios, is encouraging.

Based on their suggestions, we have made the following enhancements to our paper:

* Added Figure 2 and detailed calculations in Section 2, paragraph "Protocols for Nonlinearities," to clarify the computation of the number of AND gates (pyfb).
* Expanded Appendix A with Figure 8, showcasing PriViT's performance across different ViT model variants ranging from 5M to 80M parameters (BazG).
* Enhanced the explanation of knowledge distillation in Section 3.3, now with a mathematical equation (HTX8).
* Detailed the role of Delphi in our framework in Appendix D, Figure 16, including an estimation of its latency relative to PriViT (y916).
Refined our methodology description around PriViT R/G in Section 4.1 (pyfb).
* Added new data in Table 5 and discussed the advantages of PriViT over MPCViT, beyond latency improvements, in Section 4.2 (BazG, pyfb).
* Included a discussion of our method's limitations in the Conclusions section (BazG).
* Improved the paper's clarity and coherence by repositioning Tables 7 and 8 from the Appendix, correcting mathematical and typographical errors (BazG, pyfb).

We also provide individual responses to reviewer comments below.

---

Thanks again for the feedback, which certainly helped make the manuscript stronger.

---

### Meta-Review · Area_Chair_9QtP · 2023-12-05

**Metareview:**

After discussion, major concerns about the originality, performance and paper organization  remain. After carefully reading the paper, the review comments, the AC deemed that the paper should undergo a major revision, thus is not ready for publication in the current form.

**Justification For Why Not Higher Score:**

N/A

**Justification For Why Not Lower Score:**

N/A

---

### Decision · Program_Chairs · 2024-01-16

Reject